



# Improved RepVGG Ground-Based Cloud Image Classification with Attention Convolution

**Chaojun Shi**[1,2]**, Leile Han**[1]**, Ke Zhang**[1,2]**, Hongyin Xiang**[1,2]**, Xingkuan Li**[1]**, Zibo Su**[1]**, and Xian Zheng**[1]

[1]Department of Electronic and Communication Engineering, North China Electric Power University, Baoding 071003, China
[2]Hebei Key Laboratory of Power Internet of Things Technology, North China Electric Power University, Baoding Hebei 071003, China

**Correspondence:** Chaojun Shi (scj@ncepu.edu.cn) and Hongyin Xiang (66283880@qq.com)

**Abstract.** Clouds greatly impact the earth's radiation prediction, hydrological cycle, and climate change. Accurate automatic

recognition of cloud shape based on ground-based cloud image is helpful to analyze solar irradiance, water vapor content, and atmospheric motion, and then predict photovoltaic power, weather trends, and severe weather changes. However, the appearance of clouds is changeable and diverse, and its classification is still challenging. In recent years, convolution neural network(CNN) has made great achievements in ground-based cloud image classification. However, traditional CNN has a poor ability to associate long-distance clouds, so extracting the global features of cloud images is difficult. Therefore, a

ground-based cloud image classification method based on improved convolution neural network RepVGG and attention mechanism is proposed in this paper. Firstly, the proposed method increases the RepVGG residual branch and obtains more local detail features of cloud images through small convolution kernels. Secondly, an improved channel attention module is embedded after the residual branch fusion, which can effectively extract the global features of the cloud images. Finally, the linear classifier is used to classify the ground cloud images. In addition, the warm-up method is introduced to optimize the

learning rate in the training stage of the proposed method, and it is lightweight in the inference stage, which can avoid over-fitting and accelerate the convergence speed of the model. The proposed method in this paper is evaluated on MGCD and GRSCD ground-based cloud image datasets, and the experimental results show that the accuracy of this method reaches 98.15% and 98.07%, respectively, which exceeds other most advanced methods, and proves that CNN still has room for improvement in ground-based cloud image classification task.

# 1 Introduction

Cloud is a common and important natural phenomenon, which is water droplets, ice crystals, or aggregates of both in the atmosphere, accounting for more than 60% of the global land area (Qu et al., 2021; Gyasi and Swarnalatha, 2023; Fabel et al., 2022). Cloud analysis plays a crucial role in meteorological observation because clouds can affect the Earth's water cycle, climate change, and solar irradiance (Gorodetskaya et al., 2015; Goren et al., 2018; Zheng et al., 2019). Cloud observation

methods mainly include satellite observation (Norris et al., 2016; Zhong et al., 2017; Li et al., 2023) and ground observation



(Calbó and Sabburg, 2008; Nouri et al., 2019; Lin et al., 2023). Satellite observation refers to the distribution, movement, and change of clouds observed by high-resolution remote sensing satellites from the perspective of the outside to the inside. When observing local sky areas, its decimeter-level observation performance cannot obtain a sufficient resolution to describe the characteristics of different clouds blending in detail. Compared with satellite observation, ground-based observation
opens up a new way to monitor and understand regional sky conditions. Its equipment can not only observe local sky areas but also capture ground-based cloud images with higher resolution. The observation effect is shown in Figure 1. Typical ground-based cloud observation instruments include All-Sky Imager (ASI)(Shi et al., 2019; Cazorla et al., 2008), Total Sky Imager (TSI)(Johnson et al., 1989; Tang et al., 2021), etc. The appearance of the equipment and ground-based cloud images taken are shown in Figure 2.

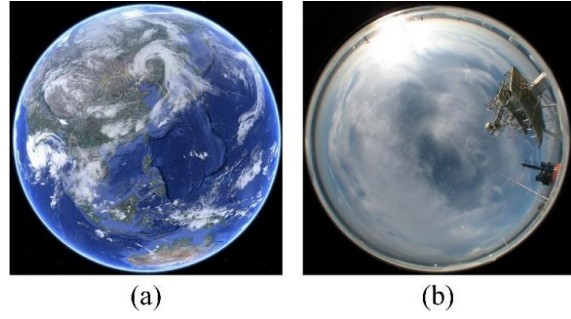

(a)  (b)


**Figure 1.** Satellite cloud and Ground-based cloud observation images. (a) Satellite cloud iamge.The picture is from the National Satellite Meteorological Center: http://www.nsmc.org.cn/ (b) Ground-based cloud image(Johnson et al., 1989).

Ground-based cloud observation can obtain more obvious cloud characteristics by observing the information at the bottom of the cloud, which is conducive to assisting the prediction of local photovoltaic power generation. Clouds play an
important role in maintaining the atmospheric radiation budget balance by suppressing short-wave and long-wave solar radiation (Taravat et al., 2015). Pv power prediction is affected by multiple factors such as cloud genus, cloud cover change, solar irradiance, and solar cell performance in local areas, among which cloud genus is an important factor affecting PV power prediction (Zhu et al., 2022). Therefore, it is of great significance to accurately obtain sky cloud information through cloud observation and then accurately classify clouds for accurate prediction of photovoltaic power generation (Alonso-
Montesinos et al., 2016). The traditional ground-based cloud observation method is mainly manual observation, which relies heavily on the experience of observers, cannot achieve standardization, and has low efficiency. Therefore, ground-based cloud automatic observation has been widely concerned by scholars. In recent years, with the development of digital image acquisition devices, many ground-based whole-sky cloud image acquisition devices have emerged at home and abroad, providing massive data support for automatic ground-based cloud observation (Pfister et al., 2003).
Ground-based cloud image classification is an important part of the foundation of automatic cloud observation and is the key to climate change and photovoltaic power prediction. The classification of ground-based cloud images mainly classifies each cloud image taken from the ground into the corresponding cloud genus by extracting cloud image features, such as cirrus, cumulus, stratus, stratus nimbus, etc. According to different cloud image feature extraction methods, the



ground-based cloud image classification method is divided into based on traditional machine learning method and based on deep learning method (Hu et al., 2018). Most of the ground-based cloud image classification methods based on traditional machine learning classify cloud images by artificially designing cloud image features, while the ground-based cloud image classification methods based on deep learning mainly classify cloud images through self-learning cloud image features of deep neural network (DNN) (Wu et al., 2019).

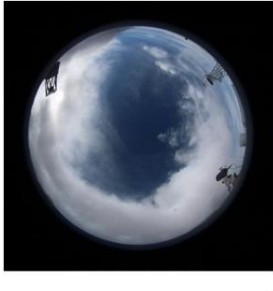 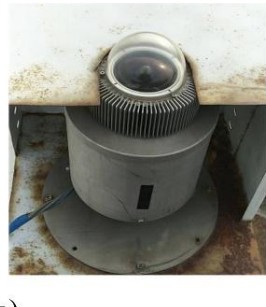 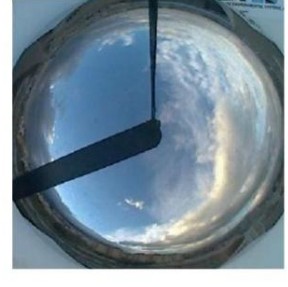 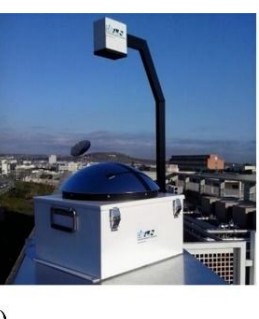

(a) (b)

**Figure 2.** Two kinds of ground-based cloud images and their observation equipment. (a) TSI ground-based cloud image and its observation equipment(Johnson et al., 1989). (b) ASI ground-based cloud image and its observation equipment(Cazorla et al., 2008).

Early ground-based cloud image classification studies relied on manual classification methods, which focused on features such as texture, structure, and color, combined with traditional machine learning methods to classify ground-based cloud images. These methods include a decision tree, K-nearest neighbor (KNN) classifier, support vector machine (SVM), etc. Singh et al. (Singh and Glennen, 2005) proposed a method for automatically training the texture function of a cloud classifier. In this method, five feature extraction methods including autocorrelation, co-occurrence matrix, edge frequency, Laws texture analysis, and original length are used respectively. Compared with other cloud classification methods, this method has the advantages of high accuracy and fast classification speed, but its classification ability for mixed clouds is insufficient. Heinle et al. (Heinle et al., 2010) described cloud images by using spectral features (mean value, standard deviation, skewness, and difference) and texture features (energy, entropy, contrast, homogeneity, and cloud cover), and combined with a KNN classifier, divided ground cloud images into seven categories. In addition, Zhuo et al. (Zhuo et al., 2014) proposed that the spatial distribution of contour lines could represent the structural information of cloud shapes, used the central description pyramid to simultaneously extract the texture and structural features of ground-based cloud images, and used SVM and KNN to classify cloud images. It can be seen that the traditional classification method of ground-based cloud images based on machine learning mainly uses hand-designed texture, structure, color, shape, and other features to extract, and obtains high-dimensional feature expression of ground-based cloud images through single feature or fusion feature. Traditional machine learning methods mostly describe the features from the perspective of digital signal analysis and mathematical statistics, but ignore the representation and interpretation of the visual features of the cloud image itself.

In recent years, under the background of cross-integration of different disciplines and artificial intelligence, the ground-based cloud image classification method based on deep learning has become a research hotspot with its superior



classification performance. Aiming at the unique characteristics of ground-based cloud images, Shi et al. (Shi et al., 2017) proposed Deep Convolutional Activations-Based Features ( DCAFs ) to classify ground-based cloud images, and the results are better than the artificially designed cloud image features. Ye et al. (Ye et al., 2017) used CNN to extract cloud image features and proposed a local pattern mining method based on ground-based cloud images to optimize the local features of

cloud images and improve the classification accuracy of cloud images. Zhang et al. (Zhang et al., 2018a) put the wake cloud as a new genus of cloud into the ground-based cloud image database for the first time, proposed a ground-based cloud image classification method based on CloudNet, and improved the classification accuracy of the ground-based cloud images. Wang et al. (Wang et al., 2020) proposed an improved CloudA network based on AlexNet. The classification accuracy on the Singapore Whole-Sky Imaging Categories ( SWIMCAT ) ground-based cloud image dataset exceeds the traditional ground-

based cloud image classification methods. Liu et al. (Liu et al., 2020b) proposed Multi-Evidence and Multi-Modal Fusion Networks (MMFN) by fusing heterogeneous features, local visual features, and multi-mode information, which significantly improves the classification accuracy of cloud images. Aiming at the problem that the traditional neural network has insufficient ability to classify the ground-based cloud images within and between genera, Zhu et al. (Zhu et al., 2022) proposed to use of an improved combined convolutional neural network to classify the cloud images, and the classification

accuracy is greatly improved compared with the traditional neural network. At the same time, Yu et al. (Yu et al., 2021) used two sub-convolutional neural networks to extract features of ground-based cloud images and used weighted sparse representation coding to classify them, which solved the problem of occlusion in multi-mode ground-based cloud image data and greatly improved the robustness of cloud images classification. Liu et al. (Liu et al., 2020a) proposed a ground-based cloud image classification method based on a graph convolution network (GCN) for the first time. However, the weight as-

signed by GCN cannot accurately reflect the importance of connection nodes, thus reducing the discrimination of aggregated cloud image features. To make up for this deficiency, Liu et al. (Liu et al., 2022) proposed a context attention network for ground-based cloud classification and publicly released a new cloud classification dataset. In addition, Liu et al. (Liu et al., 2020c) further combined CNN and GCN to propose a multimodal ground-based cloud image classification method based on heterogeneous deep feature learning. Wang et al. (Wang et al., 2021) proposed a ground-based cloud image classification

method based on Transfer Convolutional Neural Network (TCNN) by combining deep learning and transfer learning. Li et al. (Li et al., 2022) further improved the classification performance of ground-based cloud images based on the improved Vision Transformer combined with EfficientNet-CNN. The performance of the above ground-based cloud image classification method based on deep learning has been significantly improved compared with the traditional machine learning method.

CNN plays an important role in the field of target detection, image classification, and image segmentation, especially in the tasks of power line fault detection (Zhao et al., 2016), face recognition (Meng et al., 2021), and medical image segmentation (Zhang et al., 2021), and has been widely used and achieved great achievements. Ground-based cloud image classification is an emerging task in the field of image classification and has achieved rapid and considerable development based on the CNN method. However, it still has some shortcomings such as shallow network level of ground-based cloud



image classification method, limited ground-based cloud image classification performance, and small ground-based cloud image classification dataset, which cannot verify the generalization ability of large-scale ground-based cloud image classification dataset. To solve the above problems, RepVGG (Ding et al., 2021) is used in the classification task of ground-based cloud images for the first time in this paper, and a new classification method for ground-based cloud images is proposed by improving RepVGG. In this method, the ground-based cloud image is input into the CNN model and its image

features are extracted. Multi-branch convolution layer and channel attention module are used to capture local and global features of the cloud image at the same time to improve the classification performance of ground-based cloud images. The results of the subsoil cloud image classification dataset MGCD (Liu et al., 2020a) and GRSCD (Liu et al., 2020b) show that the classification performance of this method is far superior to the existing subsoil cloud image classification methods.

The main contributions of this paper are summarized as follows:

(1) This paper proposes Improved RepVGG ground-based cloud image classification with attention convolution. This method broadens the residual structure and combines the advantages of the attention mechanism, which can not only extract the global features of the cloud image, but also describe the local features of the image in detail, and maximize the complementary advantages of the two in the ground-based cloud image classification.

(2) In this paper, the Efficient Channel Attention network (ECA) is improved and introduced into the feature extraction
process of ground-based cloud images. The feature extraction method is optimized through local cross-channel interaction without dimensionality reduction. In addition, the introduction of structure reparameterization in the inference stage can reduce the complexity of the model, improve the feature extraction performance, and enhance the network's learning ability of ground-based cloud image features.

(3) The comparative experimental results on the ground-based cloud images classification dataset MGCD show that the
proposed method in this paper has the best classification accuracy. The comparative experiment is carried out on the ground-based cloud image classification dataset GRSCD to further verify the generalization ability of the ground-based cloud image classification method proposed in this paper. In addition, the proposed method introduces the warm-up method to dynamically adjust the learning rate and optimize its training process.

## 2. Methods

### 2.1 Overview of Method

This section shows the overall architecture of the proposed RepVGG-based improved classification method, as shown in Figure 3. In the CloudRVE training process, CloudRVE Block with a multi-branch topology structure is used to extract features of ground-based cloud images. The multi-branch topology structure has rich gradient information and a complex network structure, which can effectively improve the characterization ability of local feature information of ground-based

cloud images. Feature maps extracted by CloudRVE Block enter the New Efficient Channel Attention (NECA) network and learn the feature relationships between sequences to obtain the global feature representation of an image. In addition, the



warm-up method is introduced into the CloudRVE training process to dynamically optimize the learning rate and accelerate the model parameter convergence to enhance the model training effect. CloudRVE inference process uses the single branch topology structure of VGG-style (Simonyan and Zisserman, 2015), and through structural reparameterization, the multi-
branch convolutional layer and batch normalization (BN) (Ioffe and Szegedy, 2015) are converted into a 3×3 convolutional layer, increasing its inference speed. The CloudRVE training process and inference process use the linear classifier to classify the ground-based cloud images to get the final result. The specific framework parameter information of the model is shown in Table 1, where a and b are magnification factors used to control the network width. The specific contents of each part are as follows.

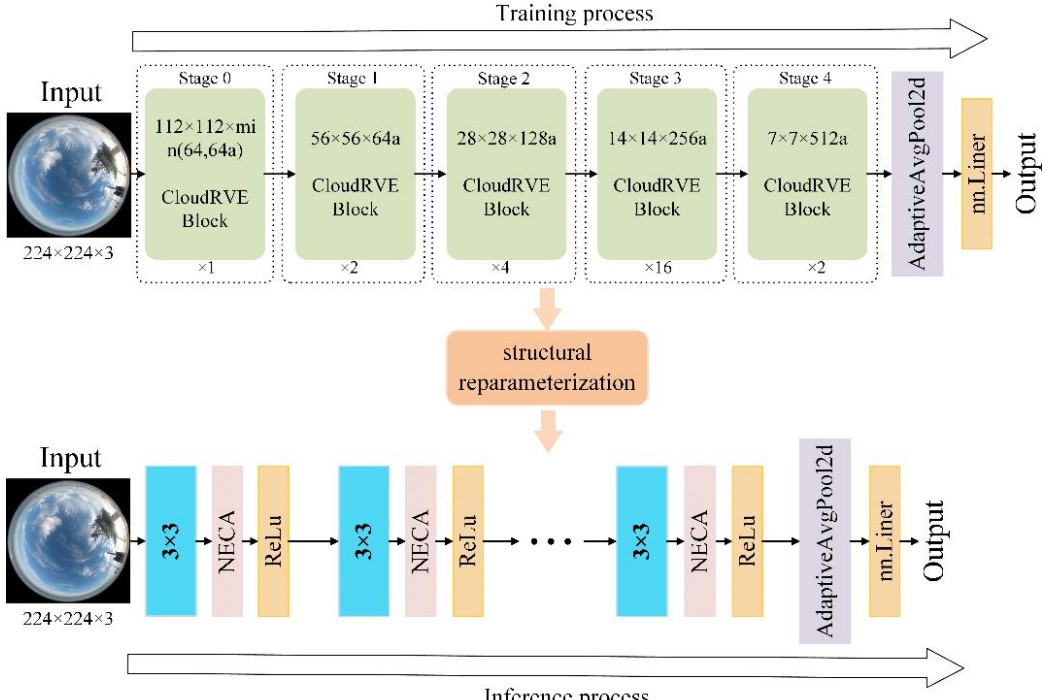


**Figure 3.** CloudRVE network framework. Ground-based cloud images come from Kiel-F datasets(Kalisch and Macke, 2008).

**Table 1. The details of CloudRVE training architecture.**

| Stage | Blocks of each stage | Output size | Output channels |
|-------|---------------------|-------------|-----------------|
| 1 | 1 | 224×224 | Min(64,64a) |
| 2 | 2 | 112×112 | 64a |
| 3 | 4 | 56×56 | 128a |
| 4 | 14 | 28×28 | 256a |
| 5 | 1 | 14×14 | 512b |



## 2.2 Broaden the CloudRVE Block of Residual Structure

In the early days, the classification of ground-based cloud images used hand-made features or shallow learning strategies,
which took a long time and had poor accuracy (Xiao et al., 2016). With the continuous development of deep learning, CNN
has been gradually applied to the task of ground-based cloud image classification and has become the mainstream method.
CNN is a deep learning model including convolution calculation, including feedforward neural network, which has
representation learning ability, similar to artificial neural network multilayer perceptron (Shi et al., 2017). In 2014, the most
representative convolution neural network VGG came out, which adopted a single-branch topology structure, greatly
improved the image processing effect and model inference speed, and became a new direction for scholars to learn and
develop. With the indepth study of the VGG, its potential in image processing is close to saturation. Scholars realize that the
VGG has some shortcomings such as simple network structure, few network layers, and large parameters, which makes it
difficult to extract high-order features of images and has limited image processing performance. Therefore, improving
network complexity and increasing the number of network layers has become a new research direction. In 2015, He
Kaiming's team at Microsoft Lab proposed ResNet (He et al., 2016), which is different from the traditional neural network
stacked by convolution layer and pooling layer. The network is stacked by residual modules, which not only increases the
complexity of the network structure and reduces the number of network parameters, but also perfectly solves the problem of
gradient disappearance or gradient explosion caused by increasing the number of network layers, which can extract abstract
image features with semantic information and effectively improve image processing performance. By improving the
complexity and depth of the network, the ResNet can train the CNN model with higher accuracy, but there are a lot of
redundancies in its residual network, which will slow down the network inference speed, increase the inference time and
reduce the accuracy of image processing results (Szegedy et al., 2015). Therefore, increasing the complexity and depth of the
network, weakening its influence on inference speed, and improving the classification effect of ground-based cloud images
have become the key contents of our research.

To improve the classification effect of the ground-based cloud images, the CloudRVE training process is composed of
CloudRVE blocks that adopt the multi-branch topology. The CloudRVE Block contains 4 branches and the improved chan-
nel attention module NECA. Its main branch contains a convolutional layer with a convolution kernel size of 3×3, which can
inspect the input images with a larger neighborhood scope and extract global features easily. Ground-based cloud images
contain abundant cloud shape and cloud amount information, while a large convolution kernel tends to ignore cloud
boundary features, resulting in inadequate feature extraction from ground-based cloud images. Therefore, the two bypass
branches of CloudRVE Block adopt the convolution layer with the convolution kernel size of 1×1, which can not only
extract fine cloud boundary features and abstract cloud cover features but also keep the output dimension consistent with the
input dimension, facilitating the multi-branch ground-based cloud image feature fusion. The third bypass branch of
CloudRVE Block adopts the Identity branch, whose purpose is to take the input as the output and change the learning
objective to the residual result approaching 0 so that the accuracy does not decline with the deepening of the network. In



addition, each branch is connected to the BN layer, not only to avoid overfitting but also to prevent gradient disappearance or explosion. The specific structure of CloudRVE Block is shown in Figure 4. The input feature maps pass through three branches with a convolutional layer and BN layer at the same time. The output obtained by the input feature maps is summed with the Identity branch and input into the NECA module to obtain the final output feature.

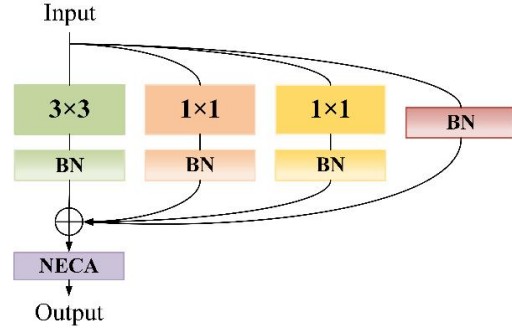

**Figure 4.** CloudRVE Block structure.

**2.3 NECA Module Focusing on Full Image Features**

The attention mechanism is to let the neural network have the information processing way to distinguish the key points and to capture the connection between global information and local information flexibly. Its purpose is to enable the model to obtain the target region that needs to be focused on, put more weight on this part, highlight significant useful features, and suppress and ignore irrelevant features. ECA module is an implementation form of channel attention mechanism, which can strengthen channel features without changing the size of the input feature maps. It adopts a local cross-channel interaction strategy without dimensionality reduction so that the 1×1 convolution layer can replace the full connection layer to learn channel attention information, which can effectively avoid the negative impact of dimensionality reduction on channel attention learning. The network performance is guaranteed and the complexity of the model is significantly reduced.

The ground-based cloud image samples can refer to Figure 2, which is taken by the all-sky imager and can cover all of the skies in this area. However, the ground-based cloud images contain not only the valid area of the whole sky but also the black invalid area. Therefore, the NECA module abandons the traditional global maximum pooling and adopts double global average pooling. The global average pooling formula is as follows:

$$\gamma_{gap} = \frac{1}{wh}\sum_{i=1,j=1}^{w,h} X_{ij\cdot} \ , X \in R^{w\times h\times c} \tag{1}$$

$$\eta_{gap} = \sigma(V_k^{gap}\gamma_{gap}) \, , V_k^{gap} \in R^{c\times c} \tag{2}$$

Where $X$ represents the input feature maps, $X'$ represents the output feature maps, and $w$, $h$, $c$ represents the width, height, and number of channels of the input feature map. The NECA module adopts a double global average pool, which can effectively improve its noise suppression ability and enhance its channel feature extraction ability, which can avoid the black invalid part of the feature calculation. The NECA module model structure is shown in Figure 5.



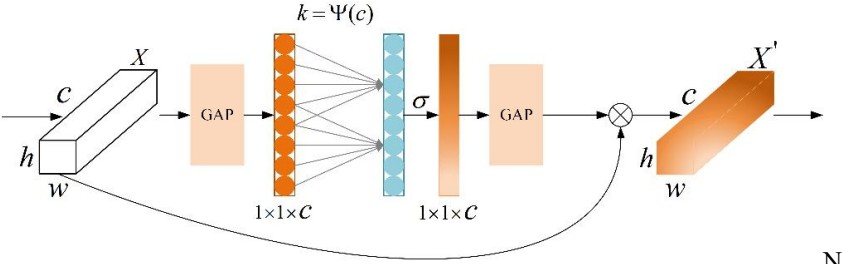

**Figure 5.** NECA model structure.

$b$ and $r$ are fixed values, and their values are set to 1 and 2, respectively. $k$ represents the convolution kernel size and has a corresponding relationship with $c$. As the network deepens, the number of channels $c$ increases by the power of 2. Therefore, $k$ should not be a fixed value, but a dynamic change and its relationship are as follows:

$$C = \phi(K) = 2^{(\gamma*k-b)} \tag{3}$$

$$K = \psi(C) = \left| \frac{log_2(c)}{r} - \frac{b}{r} \right|_{odd} \tag{4}$$

**2.4 Inference Process from Multi-Branch to Single-Branch**

The residual module is crucial to the CloudRVE training process. Its multi-branch topology can improve CloudRVE Block's ability to extract ground cloud image features and solve optimization problems such as gradient disappearance and gradient explosion caused by increasing network depth. However, the multi-branch topology will occupy more memory for the CloudRVE reasoning process, resulting in insufficient utilization of hardware computing power and slower reasoning speed. If the single-branch topology is adopted, the computing load is reduced and the inference time is saved, thus reducing memory consumption. Therefore, the single-branch topology structure is adopted in the CloudRVE inference stage, and the trained CloudRVE Block needs to be transformed into a single-branch topology model through structural reparameterization. The conversion process mainly includes the fusion of the convolutional layer and BN layer, the conversion of the BN layer into a convolutional layer, and the fusion of the multi-branch convolutional layer. We use $W_{(3)} \in R^{C_1 \times C_2 \times 3 \times 3}$ as 3×3 convolution layers, and use $C_1$, $C_2$ as input channels and output channels respectively, and use $W_{(1)} \in R^{C_1 \times C_2 \times 1 \times 1}$ as 1×1 convolution layers. In addition, we use $\mu_{(3)}, \sigma_{(3)}, \gamma_{(3)}, \beta_{(3)}$ to represent the mean value, standard deviation, learning scaling factor, and deviation of the BN layer of the main branch, and use $\mu_{(1)}, \sigma_{(1)}, \gamma_{(1)}, \beta_{(1)}$ to represent the parameters of the BN layer of the by-pass branch containing 1×1 convolution layer, and use $\mu_{(0)}, \sigma_{(0)}, \gamma_{(0)}, \beta_{(0)}$ to represent the parameters of the BN layer of the identity branch, and use $M_{(1)} \in R^{N \times C_1 \times H_1 \times W_1}$, $M_{(2)} \in R^{N \times C_2 \times H_2 \times W_2}$ to represent the input and output. The CloudRVE Block structure reparameterization calculation process is as follows:

$$\begin{aligned} M_{(2)} = &\ bn(M_{(1)} * W_{(3)}, \mu_{(3)}, \sigma_{(3)}, \gamma_{(3)}, \beta_{(3)}) + bn(M_{(1)} * W_{(1)}, \mu_{(1)}, \sigma_{(1)}, \gamma_{(1)}, \beta_{(1)}) \\ &+ bn(M_{(1)} * W_{(1)}, \mu_{(1)}, \sigma_{(1)}, \gamma_{(1)}, \beta_{(1)}) + bn(M_{(1)}, \mu_{(0)}, \sigma_{(0)}, \gamma_{(0)}, \beta_{(0)}) \end{aligned} \tag{5}$$





The input feature map is inputted into the NECA module through the 3×3 convolution layer completed by fusion. The process is shown in Figure 6.

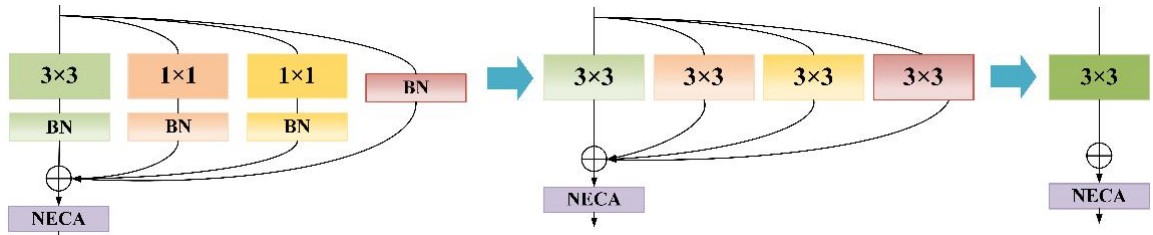

**Figure 6.** Reparameterization process of CloudRVE Block structure.

### 2.4.1 Fusion of Convolutional Layer and BN Layer

This section first describes the fusion of the main branch 3×3 convolution layer with the BN layer and then describes the transformation of the bypass branch 1×1 convolution layer into the 3×3 convolution layer and fusion with the BN layer. In the inference stage, the number of convolutional kernel channels in the convolution layer is the same as the number of channels in the input feature map, and the number of convolutional kernel channels in the output feature map is the same. The main parameters of the BN layer include mean $\mu$, variance $\sigma^2$, learning ratio factor $\gamma$, and deviation $\beta$. $\mu$ and $\sigma^2$ are

obtained statistically in the training stage, while $\gamma$ and $\beta$ are obtained by learning in the training stage. The calculation of the $i$ channel of the input BN layer is shown as follows:

$$y_i = \frac{x_i - u_i}{\sqrt{\sigma_i^2 + \varepsilon}} \times \gamma_i + \beta_i \tag{6}$$

Where $x$ is the input and $\varepsilon$ is the constant approaching 0. The calculation process of the $i$ channel input BN in the feature map is shown as follows:

$$bn(M, \mu, \sigma, \gamma, \beta)_{:,i,:,:} = \left(M_{:,i,:,:} - \mu_i\right)\frac{\gamma_i}{\sigma_i} + \beta_i = \frac{\gamma_i}{\sigma_i} M_{:,i,:,:} + \beta_i - \frac{\gamma_i}{\sigma_i}\mu_i \tag{7}$$

Where $M$ is the output feature map obtained by weighted summation of the convolution layer, input to BN layer and ignore $x$. Therefore, we can multiply $\gamma_i / \sigma_i$ to the $i$ convolution kernel of the 3×3 convolution layer:

$$W'_{i,:,:,:} = \frac{\gamma_i}{\sigma_i} W_{i,:,:,:} \tag{8}$$

$$b'_i = \beta_i - \frac{\mu_i \gamma_i}{\sigma_i} \tag{9}$$

The $i$ convolution kernel weight of the fusion of the $3 \times 3$ convolution layer and BN layer is obtained, and the specific fusion process is shown in Figures 7 and 8. The input channel $C_1$ and output channel $C_2$ are 2 and the stride is 1. In the convolution layer, the input feature map is calculated by convolution to obtain the output feature map with the number of channels 2.



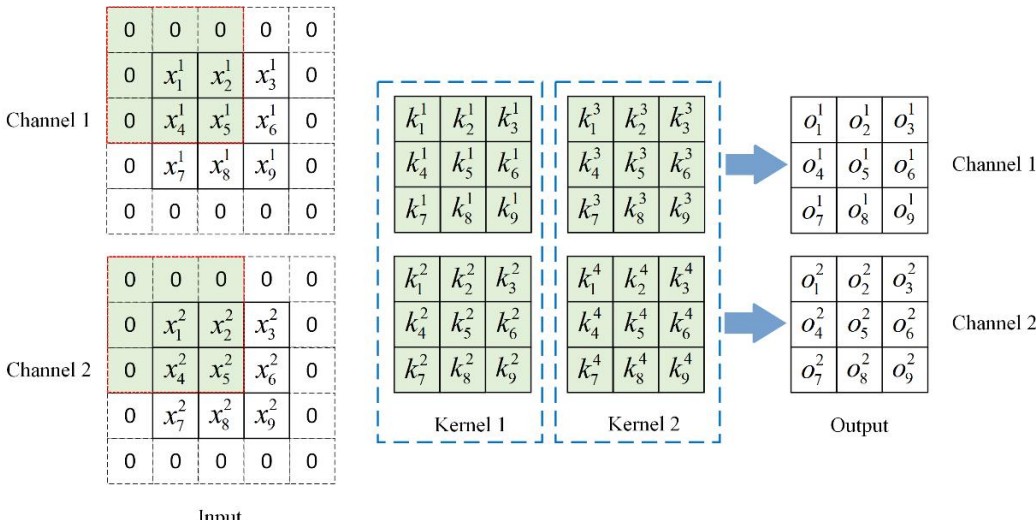

**Figure 7.** Input feature map through convolution layer process. For visualization, we assume that $C_1=C_2=2$.

In addition, to ensure that the size of the output feature map is consistent with that of the input feature map, the input feature map needs to be converted into 5 × 5 size by padding operation. The concrete convolution calculation is as follows:

$$o_1^1 = x_1^1 \cdot k_5^1 + x_2^1 \cdot k_6^1 + x_4^1 \cdot k_8^1 + x_5^1 \cdot k_9^1 + x_1^2 \cdot k_5^2 + x_2^2 \cdot k_6^2 + x_4^2 \cdot k_8^2 + x_5^2 \cdot k_9^2 \tag{10}$$

**Figure 8.** Convolutional layer output feature map through the BN layer process.

Figure 8 shows that the number of channels in the BN layer is 2, and the output feature map of the convolution layer is used as the input feature map of the BN layer. The output feature map with the number of channels being 2 is obtained by equation (2).The specific calculation process of the input feature map through the BN layer is as follows:

$$b_1 = \frac{(x_1^1 \cdot k_5^1 + x_2^1 \cdot k_6^1 + x_4^1 \cdot k_8^1 + x_5^1 \cdot k_9^1 + x_1^2 \cdot k_5^2 + x_2^2 \cdot k_6^2 + x_4^2 \cdot k_8^2 + x_5^2 \cdot k_9^2) - \mu_1}{\sqrt{\sigma^2 + \varepsilon}} \cdot \gamma_1 + \beta_1 \tag{11}$$

The result of the disassembly of equation (7) is as follows:

$$b_1 = \left(x_1^1 \cdot k_5^1 + x_2^1 \cdot k_6^1 + x_4^1 \cdot k_8^1 + x_5^1 \cdot k_9^1 + x_1^2 \cdot k_5^2 + x_2^2 \cdot k_6^2 + x_4^2 \cdot k_8^2 + x_5^2 \cdot k_9^2\right) \cdot \frac{\gamma_1}{\sqrt{\sigma^2+\varepsilon}} + \left(\beta_1 - \frac{\mu_1}{\sqrt{\sigma^2+\varepsilon}}\right) \tag{12}$$



$$c = \frac{\gamma_1}{\sqrt{\sigma^2 + \varepsilon}} \qquad\qquad d = \beta_1 - \frac{\gamma_1 \cdot \mu_1}{\sqrt{\sigma^2 + \varepsilon}} \tag{13}$$

In equation (8), $c$ and $d$ are constants and are multiplied to the first convolution kernel of the convolution layer to obtain the parameters of the first convolution kernel after the convolution layer and BN layer are fused. Other fused convolution kernel parameters are calculated similarly. The convolution layer and BN layer are fused by the bypass branch containing a 1 × 1 convolution layer. The convolution layer is first converted to 3 × 3 size by padding operation and then fused with the BN layer by repeating the above steps. The convolution layer padding process is shown in Figure 9.

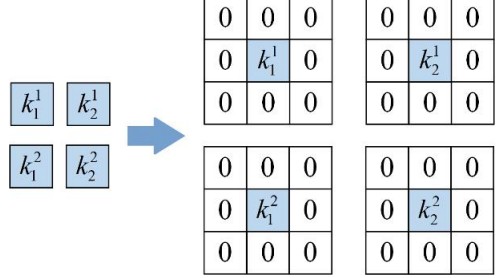

**Figure 9.** 1 × 1 convolution layer transformed into 3 × 3 convolution layer.

### 2.4.2 Convert the BN Layer to the Convolution Layer

The identity bypass branch has only a BN layer, its function is to ensure the identity mapping of the input feature map and output feature map. To realize the identical mapping between the input feature map and the output feature map in the fusion process, a 3 × 3 convolution layer with 2 convolution kernels and 2 convolution kernel channels needs to be designed. Secondly, the input feature map needs to be converted into a 5 × 5 feature map by padding operation. The specific process is shown in Figure 10. The output feature map is obtained by convolution calculation of the input feature map, and its parameters and sizes are consistent with those of the input feature map. Finally, the fusion process of the 3 × 3 convolution layer and BN layer is repeated to obtain a new 3 × 3 convolution layer.

### 2.4.3 Multi-Branch Convolution Layer Fusion

The structure reparameterization transforms each branch into a 3 × 3 convolution layer by construction and fusion, which facilitates the fusion of multi-branch convolution layers into a single-branch 3 × 3 convolution. We use $I$ to represent the input, $O$ to represent the output, $K_i$ to represent the convolution kernel weight of the i branch, $B_i$ to represent the bias of the $i$ branch. The multi-branch fusion calculation process is as follows :

$$O = (I \otimes K_1 + B_1) + (I \otimes K_2 + B_2) + (I \otimes K_3 + B_3) + I \otimes (K_1 + K_2 + K_3) + (B_1 + B_2 + B_3) \tag{14}$$





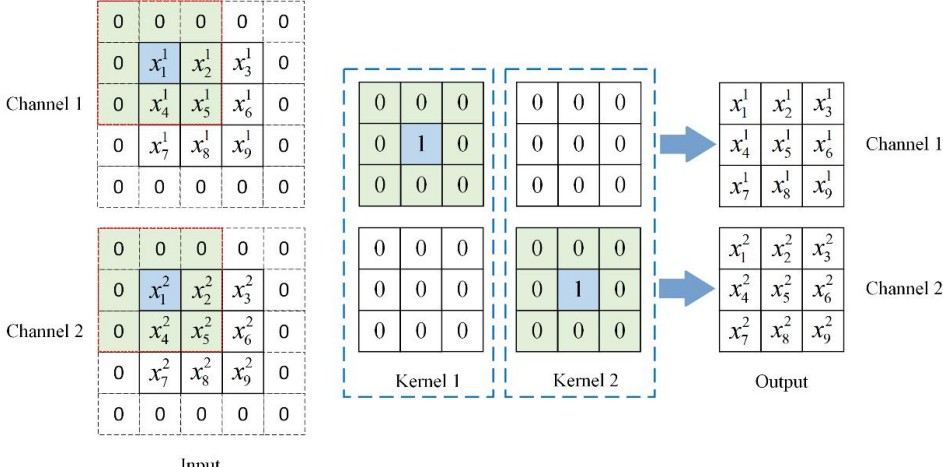

**Figure 10.** Identity branch Identity mapping process.

**2.5 Warm-Up Method**

In this paper, the warm-up method (He et al., 2019) is introduced to optimize the learning rate in the model training process, so that the learning rate is different in different model training stages. In the initial stage of model training, a small learning rate is selected, which is due to the random initialization of model weights and no prior knowledge of ground-based cloud image data, and the model will quickly adjust parameters according to the input. If a large learning rate is adopted at this time, the model will be overfitted and the prediction accuracy of the model will be affected. After training the model for some time, the learning rate linearly increases to a preset large value and the model has some prior knowledge, which can avoid overfitting and accelerate the convergence speed of the model. Finally, the model distribution is relatively stable, so it is difficult to learn new features from ground-based cloud image data, and the learning rate is linearly reduced to 0, which can keep the model stable and easily obtain local optima.

**3 Dataset and Experiment Settings**

This section introduces two kinds of ground-based cloud image classification datasets, MGCD and GRSCD, and describes the relevant experimental Settings. Section 3.1 introduces MGCD and GRSCD in detail, and Section 3.2 introduces details of experimental setting parameters and model evaluation indexes.

**3.1 Introduction of Ground-Based Cloud Image Dataset**

**3.1.1 Introduction to MGCD Dataset**

Multi-modal Ground-based Cloud image Dataset (MGCD) is the first ground-based cloud image classification dataset composed of ground-based cloud images and multi-modal information, which was collected by the School of Electronics and



Communication Engineering of Tianjin Normal University and the Meteorological Observation Center of Beijing
Meteorological Bureau of China from 2017 to 2018. There are 8000 ground-based cloud images in MGCD, and 4000
ground-based cloud images in the training set and testing set, including altocumulus(Ac), cirrus(Ci), clear sky(Cs),
cumulonimbus(Cb), cumulus(Cu), stratocumulus(Sc), and mix(Mx). In addition, cloud images with a cloud cover of less
than 10% are classified as clear sky, and each ground-based cloud image sample contains ground-based cloud images taken
at the same time and captured multimodal information. Among them, the ground-based cloud images are collected by an all-
sky camera with a fisheye lens, and its data storage format is JPEG with a resolution of 1024 × 1024 pixels; Multimodal
information is collected by weather stations, including temperature, humidity, pressure, and wind speed, and these four
elements are stored in the same vector. Figure 11 is a partial sample diagram of the MGCD dataset, and the specific
information is shown in Table 2.

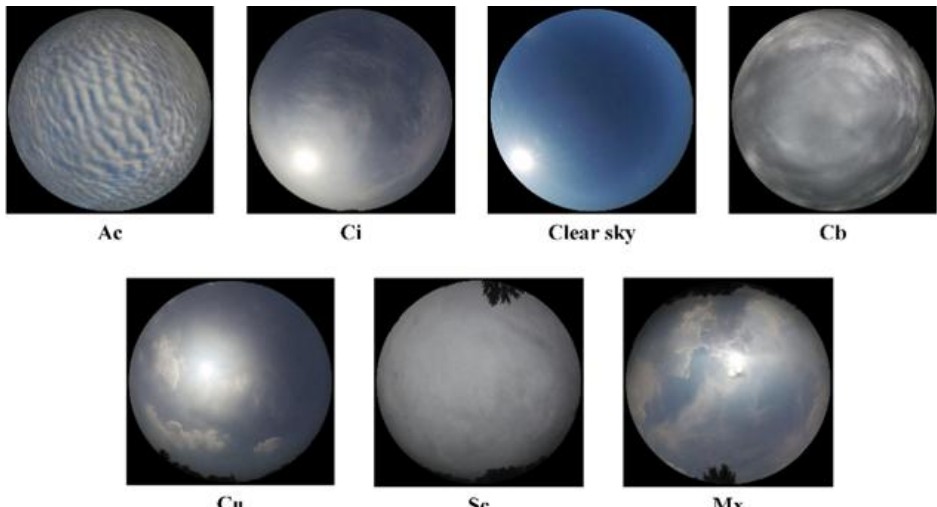

**Figure 11.** Sample legend of MGCD dataset(Liu et al., 2020a).

**Table 2.** MGCD dataset-specific information.

| No | Class | Training | Testing | Total |
|---|---|---|---|---|
| 1 | Ac | 365 | 366 | 731 |
| 2 | Ci | 662 | 661 | 1323 |
| 3 | Cs | 669 | 669 | 1338 |
| 4 | Cb | 593 | 594 | 1187 |
| 5 | Cu | 719 | 719 | 1438 |
| 6 | Sc | 482 | 481 | 963 |
| 7 | Mx | 510 | 510 | 1020 |
|  | Total | 4000 | 4000 | 8000 |

**3.1.2 Introduction to GRSCD Dataset**

Ground Remote Sensing Cloud Dataset (GRSCD) is a ground-based cloud image classification dataset composed of ground-
based cloud images and multimodal information. It was collected by the College of Electronic and Communication



Engineering of Tianjin Normal University and the Meteorological Observation Center of Beijing Meteorological
Administration of China from 2017 to 2018. There are 8000 ground-based cloud images in GRSCD, 4000 ground-based
cloud images in the training set, and 4000 ground-based cloud images in the testing set. There are seven types of clouds,
including altocumulus(Ac), cirrus(Ci), clear sky (Cs), cumulonimbus(Cb), cu-mulus(Cu), stratocumulus(Sc), and mix(Mx).
In addition, cloud images with less than 10% cloud cover are classified as clear skies. Each ground-based cloud image
sample contains ground-based cloud images taken at the same time and captured multimodal information. Among them, the
ground-based cloud images are acquired by the whole sky camera with a fisheye lens, and its data storage format is JPEG
with a resolution of 1024×1024 pixels. Multi-modal information, including temperature, humidity, pressure, and wind speed,
is collected by weather stations and stored in the same vector. Figure 12 is a partial sample of the GRSCD dataset, and the
specific information is shown in Table 3.

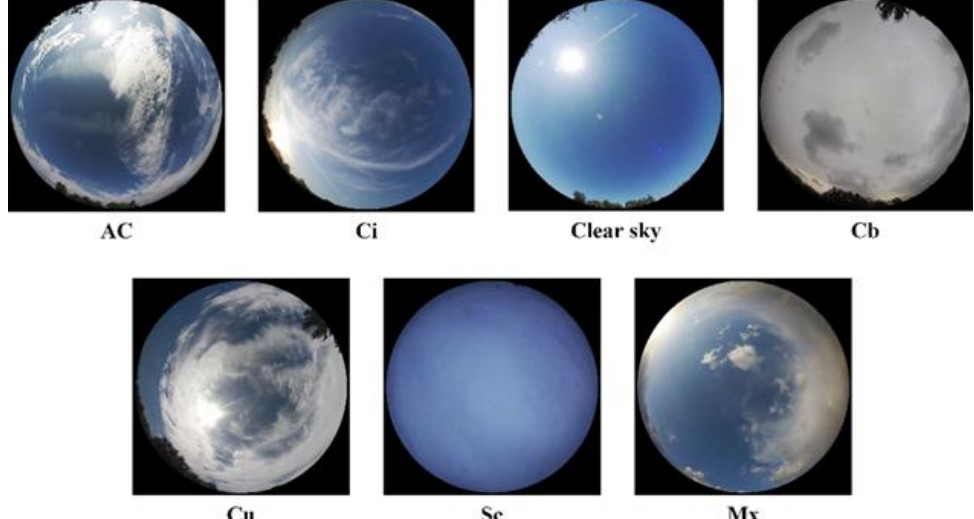

**Figure 12.** Sample legend of GRSCD dataset(Liu et al., 2020b).

**Table 3.** GRSCD dataset-specific information

| No | Class | Training | Testing | Total |
|----|-------|----------|---------|-------|
| 1 | Ac | 400 | 331 | 731 |
| 2 | Ci | 650 | 673 | 1323 |
| 3 | Cs | 650 | 688 | 1338 |
| 4 | Cb | 600 | 587 | 1187 |
| 5 | Cu | 690 | 748 | 1438 |
| 6 | Sc | 500 | 463 | 963 |
| 7 | Mx | 510 | 510 | 1020 |
|  | Total | 4000 | 4000 | 8000 |



### 3.2 Experimental Setting

#### 3.2.1 Implementation Details

All experiments in this paper adopt Python programming language and run on Intel(R) Core (TM) i9-12700K CPU @
3.60GHz. NVIDIA GeForce RTX 3090 24G Graphical Processing Unit (GPU) platform and uses Pytorch as a deep learning
framework. The CNN experiment is trained on the ground-based cloud image classification datasets MGCD and GRSCD
respectively. The number of training data accounts for 50%, the initial learning rate is set to 0.0002, Batchsize is set to 32,
and Adam optimizer (Kingma and Ba, 2015) is used to optimize all available parameters in the network. In addition, to
improve the generalization ability of the CNN model and the convergence speed of the experiment, the transfer learning
method is adopted in the training stage, and model parameters are obtained by training RepVGG with the ground-based
cloud image classification dataset made by the team and used as the weight of pre-training. CNN experiment directly trains
based on pre-training weight, which can accelerate the model convergence speed and shorten the training time, avoid the
problem of parameter overfitting, and promote the rapid gradient decline.

#### 3.2.2 Evaluation Index

To objectively evaluate the ground-based cloud image classification performance of CloudRVE and other CNN models, the
accuracy rate, accuracy rate, recall rate, and the average values of different indexes of 7 types of clouds in MGCD and
GRSCD datasets are calculated in the experiment, which is used as evaluation indexes of CNN model. The average accuracy
rate can be calculated based on positive and negative samples, and the calculation process is as follows:

$$Accuracy(Acc) = \frac{1}{n}\sum_{i=1}^{n}\frac{TP+TN}{TP+TN+FP+FN} \tag{15}$$

True Positive(TP) indicates the sample that is correctly classified as the cloud genus, True Negative(TN) indicates the
sample that is correctly classified as another cloud genus, and False Positive(FN) indicates the sample that is incorrectly
classified as the cloud genus. False Positive(FP) indicates the number of misclassified samples of other cloud genera. The
average accuracy rate and average recall rate can be expressed as:

$$Precison(Pr) = \frac{1}{n}\sum_{i=1}^{n}\frac{TP}{TP+FP} \tag{16}$$

$$Recall(Re) = \frac{1}{n}\sum_{i=1}^{n}\frac{TP}{TP+FN} \tag{17}$$

In addition, the average specificity and average F1_score are also used as evaluation indexes of the CNN model in the
experiment, and their expressions are shown as follows:

$$Specificity(TNR) = \frac{1}{n}\sum_{i=1}^{n}\frac{TN}{FP+TN} \tag{18}$$

$$F1\_score(F1) = \frac{1}{n}\sum_{i=1}^{n}\frac{2\times Pr\times Re}{Pr+Re} \tag{19}$$



## 4 Experimental Results and Discussion

### 4.1 Classification Results of Ground-Based Cloud Images

The overall classification accuracy of the CloudRVE method proposed in this paper in MGCD and GRSCD datasets and the classification results of each cloud genus are shown in Table 4 and Table 5. It can be seen that the accuracy of CloudRVE in MGCD and GRSCD datasets can reach 98.15% and 98.07%. The characteristics of the Cs in MGCD and GRSCD datasets are easy to identify, resulting in the accuracy rate, recall rate, specificity, and F1 value reaching 100%. In the MGCD dataset, the accuracy rate, recall rate, and F1 value of the other 6 cloud genera are all greater than 95.00%, and the specificity is greater than 99.50%. The accuracy and specificity of the Ci are the highest, reaching 98.64% and 99.73%, respectively. Cu has the highest recall rate and F1 value, which can reach 99.17% and 98.89%, respectively. In addition, the recall rate and F1 value of Sc and Mx are about 2.00% lower than other cloud genera, mainly because Sc and Cb in the MGCD dataset have similar characteristics, and Mx and Ci have similar characteristics, resulting in CloudRVE's ability to classify them is decreased. In the GRSCD dataset, the accuracy rate, recall rate, and F1 value of the other 6 cloud genera are all greater than 94.00%, and the specificity is greater than 99.30%. Cu has the highest accuracy, specificity, and F1 value, which can reach 99.30%, 99.85%, and 99.16%. The recall rate of Ci is the highest, reaching 99.17%. In addition, the accuracy of the Ac is only 94.24%, mainly because the Ac contains a small amount of Sc, and CloudRVE is easy to misjudge the Ac as an Sc or an Mx. Mx contains a variety of other clouds, and the images composition is complex. Cloud clusters of different can genera vary in size and shape, resulting in a lower recall rate and F1 value.

**Table 4.** Classification results of the MGCD dataset.

| Genus | Acc(%) | Pr(%) | Re(%) | TNR(%) | F1(%) |
|-------|--------|-------|-------|--------|-------|
| Cu | | 98.62 | 99.17 | 99.70 | 98.89 |
| Ac | | 97.02 | 98.08 | 99.70 | 97.55 |
| Ci | | 98.64 | 98.94 | 99.73 | 98.79 |
| Cs | 98.15 | 100.0 | 100.0 | 100.0 | 100.0 |
| Sc | | 97.26 | 95.84 | 99.63 | 96.54 |
| Cb | | 97.13 | 97.13 | 99.51 | 97.13 |
| Mx | | 97.24 | 96.67 | 99.60 | 96.95 |

**Table 5.** Classification results of the GRSCD dataset.

| Genus | Acc(%) | Pr(%) | Re(%) | TNR(%) | F1(%) |
|-------|--------|-------|-------|--------|-------|
| Cu | | 99.30 | 99.03 | 99.85 | 99.16 |
| Ac | | 94.24 | 98.63 | 99.39 | 96.39 |
| Ci | | 97.91 | 99.24 | 99.58 | 98.57 |
| Cs | 98.07 | 100.0 | 100.0 | 100.0 | 100.0 |
| Sc | | 98.10 | 96.47 | 99.74 | 97.27 |
| Cb | | 97.33 | 98.48 | 99.53 | 97.90 |
| Mx | | 97.74 | 93.33 | 99.68 | 95.49 |



Figure 13 shows the confusion matrix of MGCD and GRSCD datasets, showing CloudRVE prediction results on MGCD and GRSCD datasets. The horizontal axis represents the true cloud image classification, while the vertical axis represents the predicted cloud image classification, where the value of the diagonal element represents the correct number of cloud image classifications and the value of the offdiagonal element represents the number of cloud image classification errors. As can be seen from Figure 13(a), in the MGCD dataset, the correct classification of the Cu is the largest, while the misclassification of the cloud images mainly comes from the Sc and Mx. The reason is that the cloud base of Sc is blackened by illumination, which is easy to be confused with the Cb. In addition, the dynamic change of cloud will lead to a change in the viewpoint of the whole sky camera, thus increasing the difficulty of cloud genus identification. As can be seen from Figure 13(b), in the GRSCD dataset, the correctly classified cloud images of the same Cu have the largest number, while the incorrectly classified cloud images mainly come from the Mx and Sc. Mx cloud is a hybrid cloud, containing a variety of different cloud genera, among which Ac, Ci, and Cu account for a large proportion, so Mx is easily mistaken for Ac, Ci, or Cu. The Sc is misjudged to be the Cb, and its cloud image features are relatively similar to the Cb image features, which increased the difficulty in distinguishing.

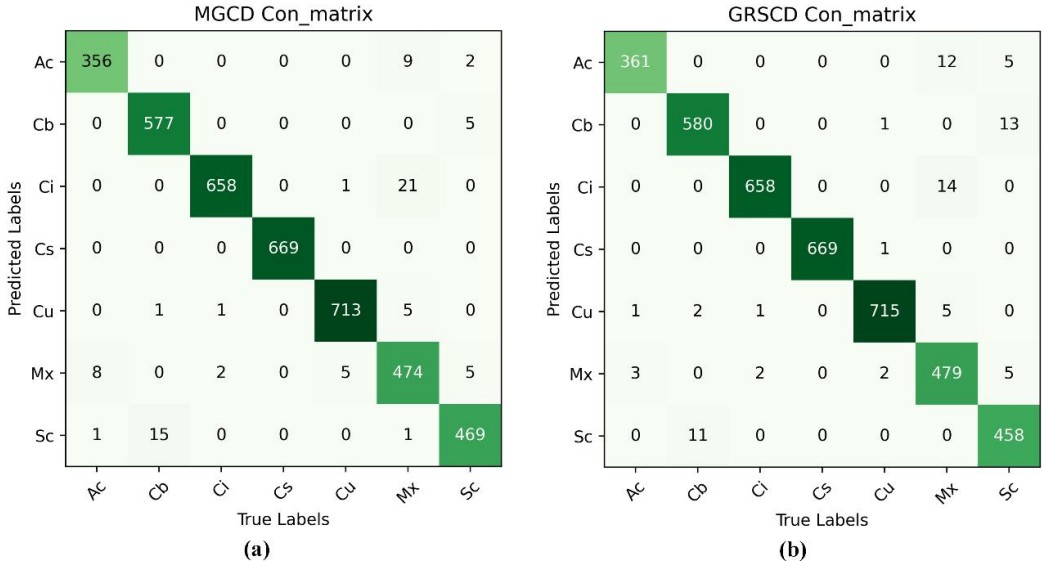

**Figure 13.** Confusion matrix images. (a)MGCD confusion matrix image. (b) GRSCD confusion matrix image.

**4.2 Ablation Experiment**

In this section, the ablation experiment is used to compare the original structure and different improvement stages of the proposed method on the MGCD and GRSCD datasets respectively, and the results are shown in Table 6. RepVGG_M is the main improved network, ECA is the attention module, CloudRVE is the combined improved network of RepVGG_M and NECA, and is the final version of the method proposed in this paper. It can be seen from the data in the table that the performance of each improvement stage of the network model is improved compared to the previous stage, which not only verifies the feasibility of extracting more cloud image detail features by adding 1×1 convolutional layer branches but also



verifies that NECA can effectively improve the noise suppression ability and enhance the channel feature extraction ability. Compared with the original network structure, the accuracy of CloudRVE in the MGCD dataset increased by 2.58%, the average accuracy rate increased by 2.68%, the average recall rate increased by 2.99%, the average specificity increased by 0.42%, and the average F1 value increased by 2.69%. In the GRSCD dataset, the accuracy rate increased by 2.65%, the
425   average accuracy rate increased by 2.81%, the average specificity increased by 0.44%, and the average F1 value increased by 2.69%. Therefore, it can be seen from the data display that the method proposed in this paper has the best performance.

**Table 6.** Results of the ablation experiment.

| Datasets | Model | Acc(%) | Pr(%) | Re(%) | TNR(%) | F1(%) |
|---|---|---|---|---|---|---|
| MGCD | RepVGG | 95.57 | 95.31 | 94.99 | 99.26 | 95.14 |
| | RepVGG_M | 95.97 | 95.65 | 95.67 | 99.33 | 95.56 |
| | RepVGG_M+ECA | 96.80 | 96.60 | 96.37 | 99.47 | 96.45 |
| | **CloudRVE** | **98.15** | **97.99** | **97.98** | **99.68** | **97.83** |
| GRSCD | RepVGG | 95.42 | 94.99 | 94.88 | 99.24 | 94.92 |
| | RepVGG_M | 95.70 | 95.46 | 95.30 | 99.29 | 95.36 |
| | RepVGG_M+ECA | 96.10 | 95.67 | 95.74 | 99.35 | 95.68 |
| | **CloudRVE** | **98.07** | **97.80** | **97.88** | **99.68** | **97.82** |

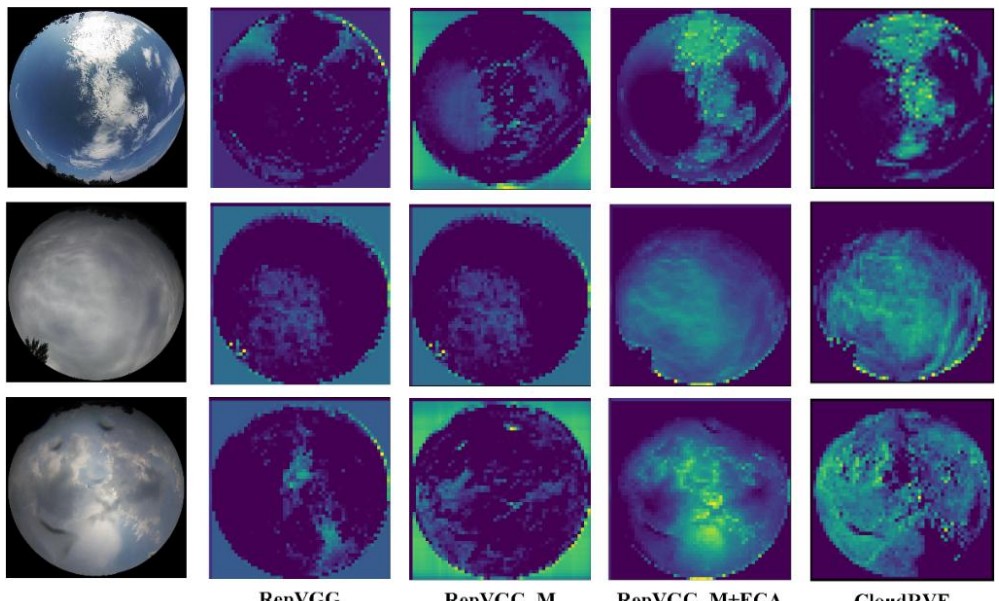

**Figure 14.** Feature extraction of different models based on MGCD(Liu et al., 2020a).

430   To visually compare the performance of the original structure and the method proposed in this paper in different improvement stages, we visualize the features by extracting the feature map of the middle layer of the network and then explain the feature extraction ability of the original structure and the method proposed in this paper in different improvement





stages, as shown in Figures 14 and 15. The method generates a rough feature map to display the important region of the predicted images through the parameter weights generated by network training, in which the brighter the region indicates the higher its importance, and the darker the region represents the sky or those that cannot be extracted. Figure 14 shows that CloudRVE has the best feature location and extraction ability by showing the feature maps of three different cloud images in the MGCD dataset. Figure 15 shows that the three cloud images of the GRSCD dataset include not only clouds and sky but also strong sunlight, which affects the classification accuracy of the model. However, it can be seen from the feature maps that CloudRVE not only has the best feature extraction ability but also has a strong ability to suppress noise such as sunlight.

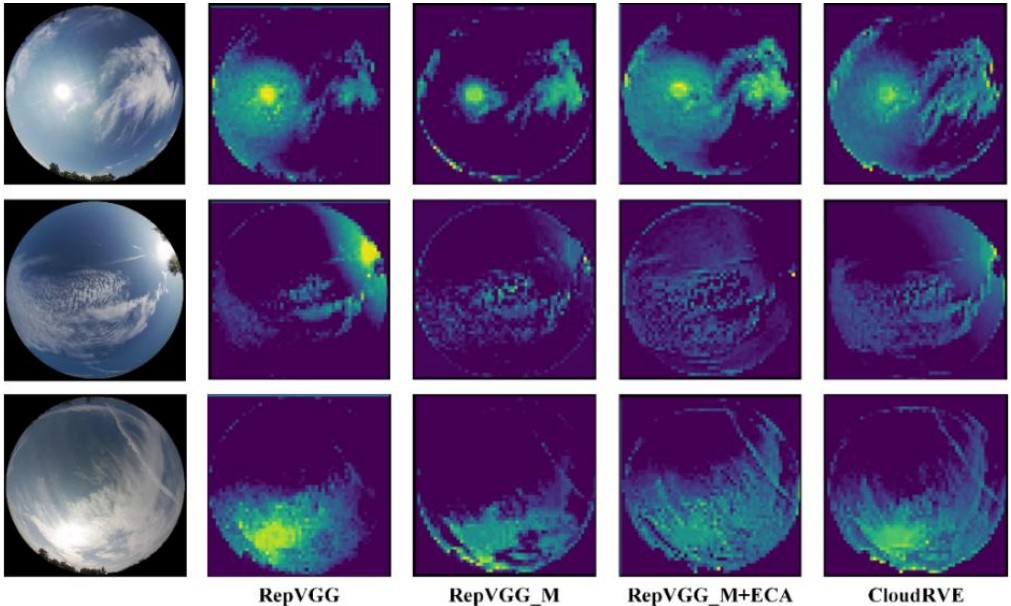

**Figure 15.** Feature extraction of different models based on GRSCD(Liu et al., 2020b).

### 4.3 Comparison of Experimental Results

To verify the superiority of the CloudRVE method, we compared it with other advanced methods, including CloudNet(Zhang et al., 2018a), CloudA(Wang et al., 2020), Eff-Siwm-T(Li et al., 2022), and other ground-based cloud image classification methods. Classic CNN models such as VGG16(Szegedy et al., 2015), ResNet50(He et al., 2016), ShuffleNet(Zhang et al., 2018b) and EfficientNet(Tan and Le, 2019). In addition, we compare it with other Transformer-based classification models such as ViT-L(Dosovitskiy et al., 2022), Swin-T(Liu et al., 2021), etc. Figure 16 and Figure 17 compare the performance of different methods by displaying the Train Accuracy and Train loss curves of MGCD and GRSCD datasets. As can be seen from the two figures, the black bold curve represents the CloudRVE method, which has the largest accuracy value, the fastest convergence rate, the smallest loss rate, and the fastest decline rate in the training stage. which further indicates that the CloudRVE method has the best classification performance of ground-based cloud images.





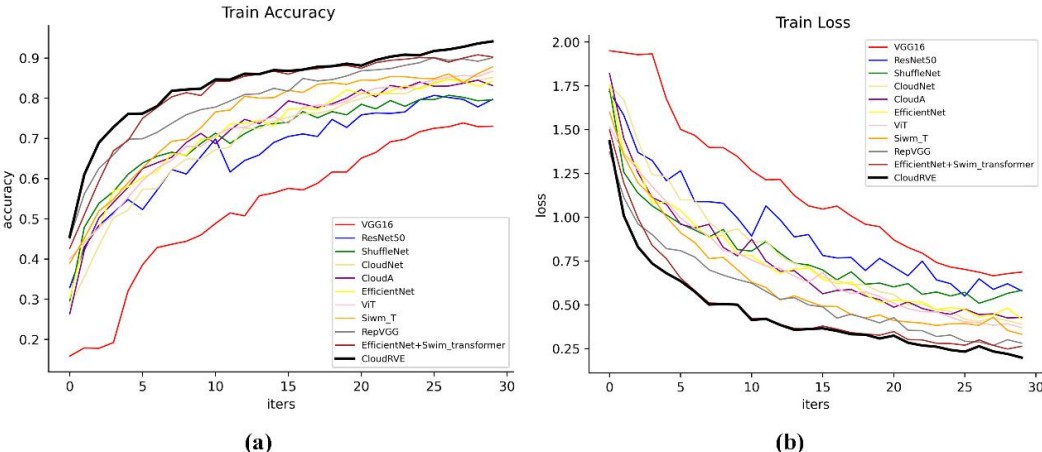

**Figure 16.** train accuracy and train loss curve of the MGCD dataset. (a) train accuracy curve of the MGCD dataset. (b) train loss curve of the MGCD dataset.

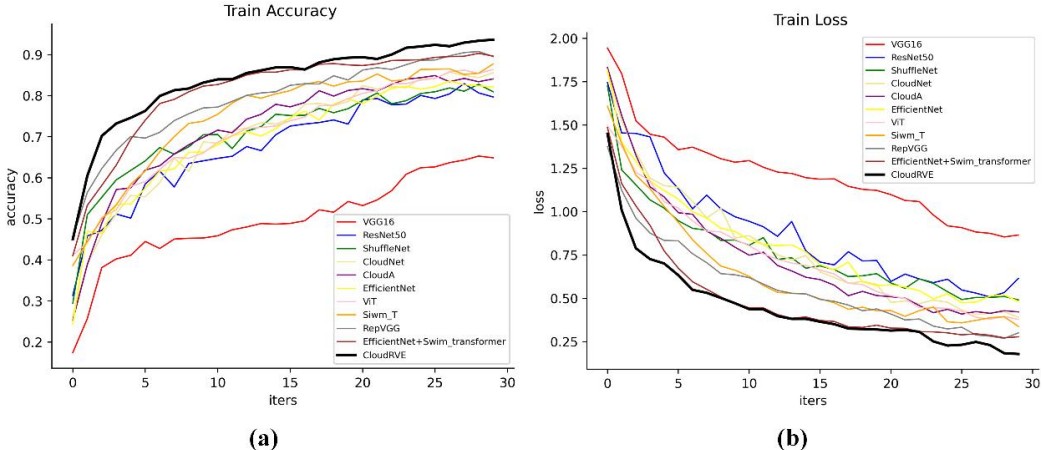

455

**Figure 17.** train accuracy and train loss curve of the GRSCD dataset. (a) train accuracy curve of the GRSCD dataset. (b) train loss curve of the GRSCD dataset.

The final comparison results of each method are shown in Table 7. It can be seen from the experimental results that RepVGG has the best performance among the CNN-based methods. The accuracy rate, accuracy rate, recall rate, specificity, and F1 value of MGCD and GRSCD datasets reached 95.57%, 95.31%, 94.99%, 99.26%, 95.14%, and 95.42%, 94.99%, 94.88%, 99.24%, 94.92%, respectively. Ground-based cloud images have more texture features and deep semantic features than other images, and more image features need to be obtained to meet the classification requirements of such images. In recent years, Transformer has been widely used for image processing tasks due to its strong feature extraction capability. Relevant scholars have improved the Transformer derivative model through continuous exploration. Among them, Eff-Siwn-T is an improvement based on Sirwn-T, and its performance on MGCD and GRSCD datasets is higher than that of the classic CNN model. The accuracy rate, accuracy rate, recall rate, specificity, and F1 value reached 96.93%, 96.73%, 96.44%, 99.49%, 96.56%, and 95.62%, 95.41%, 95.11%, 99.27%, 95.21%, respectively. Compared with Transformer and classical

460

465



networks, the results show that the proposed method can greatly improve the classification performance of ground-based cloud images. For different cloud image classification datasets, it has excellent generalization ability and strong robustness, which can play a positive role in photovoltaic power generation prediction.

**Table 7.** Comparison of experimental results.

| Methods | MGCD | | | | | GRSCD | | | | |
|---------|--------|--------|--------|--------|--------|--------|--------|--------|--------|--------|
|         | Acc(%) | Pr(%) | Re(%) | TNR(%) | F1(%) | Acc(%) | Pr(%) | Re(%) | TNR(%) | F1(%) |
| VGG-16 | 78.25 | 77.04 | 75.52 | 96.36 | 75.55 | 73.50 | 73.88 | 70.29 | 95.53 | 70.87 |
| ResNet-50 | 85.98 | 85.24 | 84.55 | 97.67 | 84.82 | 86.51 | 85.56 | 85.38 | 97.75 | 85.34 |
| ShuffleNet | 86.95 | 86.08 | 85.68 | 97.83 | 85.71 | 86.99 | 86.85 | 85.18 | 97.82 | 85.71 |
| CloudNet | 90.01 | 89.24 | 89.08 | 98.34 | 89.13 | 89.60 | 89.06 | 88.60 | 98.27 | 88.79 |
| CloudA | 89.62 | 88.78 | 88.50 | 98.28 | 88.61 | 90.03 | 89.54 | 88.71 | 98.34 | 89.03 |
| EfficientNet | 91.17 | 90.66 | 90.22 | 98.53 | 90.27 | 90.10 | 89.68 | 88.92 | 98.35 | 89.13 |
| ViT-L | 91.11 | 90.91 | 90.21 | 98.55 | 90.40 | 90.98 | 90.49 | 90.33 | 98.50 | 90.39 |
| Siwn-T | 92.87 | 92.44 | 91.63 | 98.63 | 91.76 | 93.55 | 93.22 | 92.87 | 98.93 | 92.71 |
| RepVGG | 95.57 | 95.31 | 94.99 | 99.26 | 95.14 | 95.42 | 94.99 | 94.88 | 99.24 | 94.92 |
| Eff-Siwn-T | 96.93 | 96.73 | 96.44 | 99.49 | 96.56 | 95.62 | 95.41 | 95.11 | 99.27 | 95.21 |
| **CloudRVE** | **98.15** | **97.99** | **97.98** | **99.68** | **97.83** | **98.07** | **97.80** | **97.88** | **99.68** | **97.82** |

**5 Conclusion**

This paper proposes a classification method for ground-based cloud images based on the improved network. Specifically, by improving the training stage structure of the network, broadening the RepVGG residual structure, and adding 1×1 convolutional layer branches to each block, the gradient information of the topology structure can be increased and the ability of the improved network to represent the boundary features of the cloud image can be improved. In addition, the NECA module is embedded after multi-branch fusion to learn the feature relationship between sequences, improve the network cross-channel interaction ability, and extract the best global features of cloud images. We evaluate the performance of the proposed method on MGCD and GRSCD ground-based cloud image datasets and compare it with other advanced methods, our method can achieve the highest accuracy of 98.15% and 98.07%, respectively. In recent years, the capacity of ground-based cloud image datasets has been expanding contin-uously, and the classification of ground-based cloud images by convolutional neural networks has reached the bottleneck. Transformer has gradually attracted scholars' attention and has been applied in image processing, as a powerful deep neural network for processing sequences. However, Transformer has received little attention in the field of ground-based cloud image classification. In addition, cloud classification is only based on ground-based cloud image features, but many physical features can provide the basis for cloud classification, such as height, thickness, etc. In the future, we will consider combining CNN to study and improve the Transformer model and use cloud height, cloud thickness, and other parameters in ground-based cloud image classification to improve model performance.



*Author Contributions.* LH performed the experiments and wrote the paper. CS, KZ, HX analyzed the data and designed the experiments. CS conceived the method and reviewed the paper. XL, ZS and XZ reviewed the paper and gave constructive suggestions.

*Financial support.* This research was funded by the National Science Foundation of China (NSFC) under Grant NO.
62076093 and NO. 62206095 and through the Fundamental Research Funds for the Central Uni-versities under Grant NO. 2022MS078 and NO. 2020MS099.

*Data Availability Statement.* The MGCD dataset was accessed from https://github.com/shuangliutjnu/Multimodal-Ground-based-Cloud-Database. The GRSCD dataset was accessed from https://github.com/shuangliutjnu/TJNU-Ground-based-
Remote-Sensing-Cloud-Database.

*Acknowledgments.* We would like to thank Professor Liu Shuang of Tianjin Normal University for providing the support of ground-based cloud image classification datasets and Student Meng Ru-oxuan from Guangxi Normal University for her contribution to this paper.

*Competing interest.* The authors declare that they have no conflict of interest.

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
