# Peer review of "Improved RepVGG Ground-Based Cloud Image Classification with Attention Convolution"

_EGUsphere, 2023_

## Referee Comment (RC1)

This manuscript combined the RepVGG and attention mechanism for cloud image classification. The idea is interesting and the result is basically satisfactory. However, some other problems in the manuscript are still concerned in the following:

1. The visual results of cloud image classification are suggested to be shown and compared with different methods in the experiments.
2. Compared with other methods, how is the parameter amount of the proposed method?
3. The organization of this manuscript should be added to the end of the introduction.
4. The language should be polished by an English native speaker.
5. Cloud cover has two sides. On one hand, it is helpful to analyze solar irradiance as the authors said. On the other hand, it hinders the observation on the ground, as described in "Cloud removal from satellite images using spatiotemporal generator networks", "Bishift networks for thick cloud removal with multitemporal remote sensing images". This point should be stated.
6. The type setting should be checked.

---

## Author Response (AR1)

**Final Author's Response**

First of all, we would like to thank the Associate Editor for the precious time and great efforts on reviewing the manuscript. At the same time, we also really appreciate the Anonymous Referees for the objective and pertinent comments, which will help the authors to improve the manuscript. According to the reviewing comments, we have revised the manuscript and our point-by-point responses (in green) to the comments (in black) are given. The modification made in the manuscript is presented in blue.

Best Regards,

Chaojun Shi, Leile Han, Ke Zhang, Hongyin Xiang, Xingkuan Li, Zibo Su, Xian Zheng
* * *
**Response to Anonymous Referee #1**

This manuscript combined the RepVGG and attention mechanism for cloud image classification. The idea is interesting and the result is basically satisfactory.

✓ **Author's response:** We appreciate referee the concise summary for our manuscript.

**Few comments:**

1. The visual results of cloud image classification are suggested to be shown and compared with different methods in the experiments.

✓ **Author's response:** We are extremely grateful to reviewers for their suggestion. We have shown visual results of cloud image classification in our manuscript's section 4.3 Comparison of Experimental Results, and compared it with different methods. As follows:

✓ **Modifications: Figure 18: Feature extraction of different methods based on MGCD, (a) Original (Liu et al., 2020a); (b)VGG-16; (c) ResNet-50; (d) ShuffleNet; (e) CloudNet; (f) CloudA; (g) EfficientNet; (h) ViT-L; (i) Swin-T; (j) RepVGG; (k) Eff-Swin-T; (l) CloudRVE**

[Figure]

✓ **Figure 19: Feature extraction of different methods based on GRSCD: (a) Original (Liu et al., 2020b); (b)VGG-16; (c)ResNet-50; (d) ShuffleNet; (e) CloudNet; (f) CloudA; (g) EfficientNet; (h) ViT-L; (i) Swin-T; (j) RepVGG; (k) Eff-Swin-T; (l) CloudRVE**

[Figure]

✓ **Page 24-25 Line 514-526:** In order to provide a more intuitive display of the advantages of CloudRVE over other advanced methods, we extracted the features of the intermediate layers of different methods to generate the ground cloud feature maps for the building foundation, demonstrating the strong feature extraction capabilities of CloudRVE and proving its superiority, as shown in Figures 18 and 19. Feature extraction was achieved by generating rough feature

10 maps through network training with parameter weights to highlight the important regions of predicted images. The light colored regions represent the important features, while the dark colored regions represent the sky or unsuccessfully extracted features. Figure 18(b-i) shows the feature maps of different ground cloud classification methods based on

MGCD dataset to demonstrate the CloudRVE capability to extract more extensive and comprehensive cloud features and suppress the black regions and sunlight, further illustrating the best feature localization and extraction capability of CloudRVE. Figure 19(b-i) shows the feature maps of different ground cloud classification methods based on GRSCD dataset to demonstrate that the cloud feature extracted by CloudRVE covers the effective area in Figure 19(a) with the best coverage and the best suppression of the sunlight, further proving that CloudRVE has the best feature localization and extraction capabilities.

2. Compared with other methods, how is the parameter amount of the proposed method?

✓ **Author's response:** We deeply appreciate the additional suggestions provided by the reviewers on the experimental results. We have added a comparison of methods model sizes in Section 4.3 Comparison of Experimental Results in our paper. Although our method has not been the smallest model size, our method has the highest accuracy. Here are the additional experimental results:

✓ **Modifications: Page23 Line 499-504:** The space complexities of CloudRVE and ten alternative methods are summarized and compared in Table 8. It can be seen from the table that CloudRVE had a spatial complexity of 105.17 Mb, which is in line with the spatial complexity of Swin-T and Eff-Swin-T, and far less than the spatial complexity of ViT-L. The spatial complexity of CloudRVE exceeded that of RepVGG by three times, achieving the best ground cloud image classification performance. Thus, CloudRVE achieved excellent ground cloud image classification performance at the expense of higher spatial complexity.

✓ **Table 8:**

Table 8. Space complexity of the proposed and ten alternative methods.

| Method | Space complexity (Mb) |
|---|---|
| VGG-16 | 512.28 |
| ResNet-50 | 90.03 |
| ShuffleNet | 4.93 |
| CloudNet | 153.36 |
| CloudA | 87.57 |
| EfficientNet | 15.61 |
| ViT-L | 327.37 |
| Swin-T | 105.28 |
| RepVGG | 30.10 |
| Eff-Swin-T | 105.24 |
| **CloudRVE** | **105.17** |

3. The organization of this manuscript should be added to the end of the introduction.

✓ **Author's response:** We are extremely grateful to the reviewer for the suggestion. We have added the organization of this manuscript into the introduction of the manuscript in particular, as outlined in the following:

✓ **Modifications: Page 5-6 Line 153-157:** The rest of this paper is organized as follows. Section 2 elaborates on the structure and composition of the proposed CloudRVE method for classifying ground cloud images. Section 3 briefly introduces the ground cloud image classification datasets used in this paper and the model evaluation indices. Section 4 provides the experimental results and discusses the feasibility and effectiveness of the proposed method. Finally, Section 5 concludes the study and outlines future research directions and practical application of the research results.

4. The language should be polished by an English native speaker.

✓ **Author's response:** We sincerely appreciate the reviewer's suggestion. We have invited experts native to English to polish the manuscript according to the reviewer's suggestions, and the results will be reflected in the subsequently uploaded manuscripts.

5. Cloud cover has two sides. On one hand, it is helpful to analyze solar irradiance as the authors said. On the other hand, it hinders the observation on the ground, as described in "Cloud removal from satellite images using spatiotemporal generator networks", "Bishift networks for thick cloud removal with multitemporal remote sensing images". This point should be stated.

✓ **Author's response:** As reviewers pointed out, cloud cover has a dual nature. On the one hand, it helps analyzing solar radiation. On the other hand, it hinders the observation on the ground. Considering the comments from reviewers, we modified the manuscript and cited two related papers, "Cloud removal from satellite images using spatiotemporal generator networks" and "Bishift networks for thick cloud removal with multitemporal remote sensing images".

✓ **Modifications: Page 2 Line 35-38:** Satellite observation refers to the distribution, movement, and change of clouds observed by high-resolution remote sensing satellites from the perspective of the outside to the inside. When observing local sky areas, its decimeter-level observation performance cannot obtain a sufficient resolution to describe the characteristics of different clouds blending in detail (Long et al., 2023; Sarukkai et al., 2020).

6. The type setting should be checked.

✓ **Author's response:** We are sorry that we have encountered formatting issues in the manuscript. We have carefully checked and corrected the formatting issues in the manuscript according to the template provided by Atmospheric Measurement Techniques official website, with the results as follows:

✓ **Modifications: Fulltext:** We have modified the first line indentation from 2 characters to 1 characters.

✓ **Equation (1)-(19):** The formulas in our manuscript were modified to ensure the sizes of the parameters are kept consistent with those in the manuscript text and the numbers are right-aligned.

✓ **The reference paper entry style:Fulltext:** We have modified the citation format of the literature in the text. For instance, Page 4-5 Line 116,136-137, Liu et al. published three papers in 2020, we use 2020a, 2020b and 2020c to distinguish them.
* * *
**Response to Anonymous Referee #2**

Nevertheless, the paper is in general technically sound, and could be suited for publication in Atmospheric Measurement Techniques journal provided that it is previously improved by considering some suggestions and applying some technical corrections.

✓ **Author's response:** We appreciate the reviewer's brevity in summarizing our manuscript. We will carefully consider the reviewer's suggestions and make necessary revisions to address the issues raised. In addition, we made modifications to the revised manuscript based on the changes made in terms of the number of lines. We have adjusted the positioning of each issue raised by the reviewer in the article. For example: lines 35-36.

**Few comments:**

**L.** 35. What do you mean by "its decimeter-level observation"?

✓ **Author's response:** Its decimeter-level observation refers to the accuracy of satellite observation of clouds. We revised the entire sentence. As follows:

✓ **Modifications: line 35-36:** When observing local sky regions, satellite observations have low performance and are unable to obtain sufficient resolution to describe the characteristics of different cloud layers in detail.

**L. 39.** What do you mean by "its equipment"?

**L. 39 and Fig. 1.** I would say that this is not necessary. Everyone knows that the view from the above is different of the view from below. The differences in detail and area observed are also quite well known. Moreover, there are other satellites that give much more detail of clouds, despite the image is never as detailed as from the ground.

✓ **Author's response:** We are extremely grateful to the reviewer for the suggestion. We strongly agree with the reviewer's suggestion, and deleted this unnecessary part.

**L. 39.** Johnson 1989 is indeed a pioneer work, but it's not about the TSI, but about an original prototype of the"whole sky camera"(WSI). In addition, I would say that Shields is not a coauthor of that report. If you want a reference for the TSI, you can use Long et al 2006: Retrieving Cloud Characteristics from Ground-Based Daytime Color All-Sky Images; C. N. Long, J. M. Sabburg, J. Calbó, D. Pagès; Journal of Atmospheric and Oceanic Technology vol. 23, 5(2006) pp: 633-652.

✓ **Author's response:** We are extremely grateful to express our sincere gratitude to the reviewer for their helpful feedback.We have corrected the reference citation.

**L. 46.** Taravat et al 2015 is a too specific reference for a so general statement, which may be found in atmospheric radiaton textbooks of review papers. Moreover "by suppressing short-wave and long-wave solar radiation" is note quite precise. First, other wording (absorbing, scattering,…" could be used; second long-wave solar radiation sound strange, as I think you refer to long-wave (infrared) radiation which is emitted by the ground (and clouds) not to solar radiation as such. Please clarify.

✓ **Author's response:** We sincerely appreciate the reviewer's suggestion. We revised the entire sentence. As follows:

✓ **Modifications: line 46-48:** Clouds play an important role in maintaining the atmospheric radiation budget balance by absrbing short-wave and the ground not to solar radiation.

**L. 53.** I would use"visual"instead of "manual" observations.

**L. 54.** What do you mean by "low efficiency"?

✓ **Author's response:** We sincerely appreciate the reviewer's suggestion. In addition, we believe that the efficiency of traditional ground-based cloud observation methods is low and the recognition accuracy is not high. After careful consideration, we believe that this sentence is redundant.We revised the entire sentence. As follows:

✓ **Modifications: line 52-54:** The traditional ground-based cloud observation method is mainly visual observation, which relies heavily on the experience of observers, cannot achieve standardization.

L. 56. "Home and abroad" sounds strange in a science paper. Use "worldwide" instead.

✓ **Author's response:** We sincerely appreciate the reviewer's suggestion. We have replaced "at home and abroad" with "the world". As follows:

5 ✓ **Modifications: line 54-57:** In recent years, with the development of digital image acquisition devices, many ground-based whole-sky cloud image acquisition devices have emerged the world, providing massive data support for automatic ground-based cloud observation.

L. 59-60. This sentence is repetitive.

10 ✓ **Author's response:** We express our heartfelt gratitude for the reviewer's kindly reminder. We have deleted this sentence.

L. 61. "stratus nimbus" is not a cloud genera. It could be "nimbostratus"

✓ **Author's response:** We apologize for using the wrong English word. We have replaced "stratus nimbus" with 15 "nimbostratus". As follows:

✓ **Modifications: line 59-61:** The classification of ground-based cloud images mainly classifies each cloud image taken from the ground into the corresponding cloud genus by extracting cloud image features, such as cirrus, cumulus, stratus, nimbostratus, etc.

20 L. 63. There are studies that also used feature extraction before Hu et al 2018.

✓ **Author's response:** We sincerely appreciate the reviewer's reminder. We are aware that prior to Hu et al. 2018, there were also studies that used feature extraction. On one hand, we consider Hu et al.'s paper to be more representative. On the other hand, we try to cite paper within the last 5 years. After considering both factors, we have chosen to cite this paper. In addition, we have added two classic papers prior to Hu et al. 2018 that focus on using feature extraction for 25 research.

✓ **Modifications:**

✓ [1] Krizhevsky, A., Sutskever, I., and Hinton, G. E.: ImageNet classification with deep convolutional neural networks, Commun. ACM, 60, 84–90, https://doi.org/10.1145/3065386, 2017.

✓ [2] Simonyan, K. and Zisserman, A.: Very Deep Convolutional Networks for Large-Scale Image Recognition, 30 https://doi.org/10.48550/arXiv.1409.1556, 10 April 2015.

Figure 1. I would say is the other way around (a/b). For sure, images in (b) are not from Cazorla et al 2008.

5 ✓ **Author's response:** We sincerely appreciate the reviewer's reminder. We have corrected the order of a/b and added referenced literature. As follows:

✓ **Modifications: Figure 1:** Two kinds of ground-based cloud images and their observation equipment: (a) ASI ground-based cloud image and its observation equipment (Cazorla et al., 2008; Shi et al., 2019).(b) TSI ground-based cloud image and its observation equipment (Long et al., 2006);

10 L. 92-99. Explain in few words what it is CloudNet, CloudA, AlexNet.

✓ **Author's response:** We sincerely appreciate the reviewer's suggestion. We revised the entire sentence. As follows:

✓ **Modifications: line 92-99:** Zhang et al. (2018a) put the wake cloud as a new genus of cloud into the ground-based cloud image database for the first time, proposed a simple convolutional neural network model called CloudNet, and applied it to the ground-based cloud image classification task, effectively improving the accuracy of ground-based
15 cloud image classification. More recently, (Wang et al., 2020) proposed the CloudA network, an optimized iteration of the AlexNet convolutional neural network, which reduces the number of parameters through a simplified network architecture. The classification accuracy on the Singapore Whole-Sky Imaging Categories (SWIMCAT) ground-based cloud image dataset exceeded the traditional ground-based cloud image classification methods.

20 L. 116-118. Please rewrite and clarify. This is the result of the present study? Or is like a summary of the previous paragraph?

✓ **Author's response:** We sincerely appreciate the reviewer's reminder. This sentence is a summary of the previous paragraph, and we have rewritten it. As follows:

✓ **Modifications: line 116-118:** The performance of the above-mentioned ground-based cloud image classification methods based on deep learning has significantly improved compared to traditional machine learning methods.

25

L. 131-133. What does "subsoil" cloud image classification mean? Are you anticipating a result of your study in the introduction section?

✓ **Author's response:** We sincerely appreciate the reviewer's reminder. We deleted "subsoil" and the predictive research results to make the sentence more formal. As follows:

✓ **Modifications: line 132-133:** The method's application to the multi-modal ground-based cloud dataset named MGCD (Liu et al., 2020a) and ground-based remote sensing cloud database (GRSCD) (Liu et al., 2020b) .

Figure 2 and Table 1. Why stages go from 1-5 in Table 1 and from 0-4 in fig. 3?

5 ✓ **Author's response:** We sincerely appreciate the reviewer's reminder. We have corrected the error in this Table.

L. 175-178. These sentences are a repetition of introduction.

✓ **Author's response:** We sincerely appreciate the reviewer's reminder. We have deleted this part of the content.

L. 338-340. "each ground-based cloud image sample contains ground-based cloud images taken at the same time" Please rewrite or explain.

✓ **Author's response:** We sincerely appreciate the reviewer's suggestion. We have rewritten this sentence.

✓ **Modifications: line 339-341:** In addition, cloud images with a cloud cover of less than 10% are classified as clear sky,
15 and each sample contains a captured ground cloud iamge and a set of multimodal cloud information.

Section 3.1.2. Apparently, the text is exactly the same as in section 3.1.1. Please do not repeat and focus on the differences between both datasets. Explain for example if the two datasets contain subsets of images which are the same or if, contrarily, they are totally different.

20 ✓ **Author's response:** We sincerely appreciate the reviewer's suggestion. We have rewritten Section 3.1.2 based on the reviewer's suggestion.

✓ **Modifications: line 349-360:** Ground remote sensing cloud dataset (GRSCD) is a ground-based cloud image classification dataset composed of ground-based cloud images and multimodal information. It was collected by the College of Electronic and Communication Engineering of Tianjin Normal University and the Meteorological
25 Observation Center of Beijing Meteorological Administration of China from 2017 to 2018. The total number of ground-based cloud images in GRSCD is consistent with MGCD, with a training set and a testing set each accounting for 50%, including 7 types of clouds: altostratus (Ac), cirrus(Ci), clear sky(Cl), cumulonimbus(Cb), cumulus(Cu), stratocumulus(Sc), and mix(Mx). Among them, the features of cumulonimbus and stratocumulus in MGCD are not distinct and easy to confuse; the features of altostratus and cumulus in GRSCD are not distinct and easy to confuse. In
30 addition, each sample contains a ground-based cloud image and a set of multi-modal cloud information, and cloud

images with cloud cover not exceeding 10% are classified as clear sky. Figure 11 depicts a partial sample of the GRSCD dataset. The specific data are listed in Table 3.

Explain how the "true" classification has been established (visual inspection of images?)

L. 383-387. Explain better, and use correct wording (False Positive is repeated). One may think that a sample is either classified in the correct genus or not. So it's not clear how do you have 4 options.

✓ **Author's response:** We sincerely appreciate the reviewer for raising the issue. We explained and revised it.Image classification is based on the process of pattern recognition, which involves analyzing various image features to select feature parameters, dividing the feature space into non-overlapping subspaces, and assigning each feature in the image to a specific subspace, thereby achieving classification. Cloud image classification is a problem of classifying cloud image content, which utilizes a computer to perform quantitative analysis on cloud images and classify them into several categories, replacing visual judgment by humans. In addition, TP, TN, FP and FN parameters are important parameters commonly used in the field of image classification. We reinterpreted it. as follows:

✓ **Modifications: line 384-387:** TP (True Positive) parameter is the number of correctly classified samples for a specific genus, TN (True Negative) parameter is the number of correctly classified samples for the remaining genus, and FN (False Negative) parameter is the number of misclassified samples for a specific class genus. FP (False Positive) parameter is the number of misclassified samples for the remaining classes genera.

Please do not use "Cs" for clear sky. This is confusing as Cs means "cirrostratus", which is another cloud genus. For clear sky you may use CS (uppercase) or Cl.

✓ **Author's response:** We sincerely appreciate the reviewer's suggestion. We chose to use "Cl" to represent clear sky and revised the entire text.

L. 379. "accuracy rate" is repeated.

✓ **Author's response:** We sincerely appreciate the reviewer's reminder. We deleted duplicate content.

Is there an index for each genus? What does "n" mean? Should TP, TN, … carry an "i" subindex? Are accuracy, precision,… overall indexes or they correspond to each cloud genus?

✓ **Author's response:** We sincerely appreciate the reviewer's suggestion. We have revised and improved the formula and this part of the content.

✓ We added the subscript i to the formula to indicate indexing for different cloud types.

✓ "n": Express the number of cloud genus. We added an explanation of "n" in the text.

L. 387. "precision" instead of "accuracy"?

✓ **Author's response:** We sincerely appreciate the reviewer's reminder. We replaced "accuracy" with "precision".

Eq. (19). If Pr and Re are already totals (sums) I don't understand what are you summing to obtain F1.

✓ **Author's response:** We sincerely appreciate the reviewer's reminder. We have modified Eq. (19).

✓ **Modifications:**

$$F1\_score(F1) = \frac{2 \times Pr \times Re}{Pr + Re}, \overline{F1\_score}(\overline{F1}) = \frac{1}{n}\sum_{i=1}^{n} \frac{2 \times Pr_i \times Re_i}{Pr_i + Re_i}$$

Table 4, 5. Why there is a single Accuracy value but values for each cloud genera for the other indexes?

✓ **Author's response:** We sincerely appreciate the reviewer's question. Due to the small accuracy difference between each cloud genus in the two datasets, there is no comparison between them. Therefore, we directly use the average accuracy.

L. 403. The correct classification of the Cu is the largest because in the datasets the number of Cu images is the largest too. I mean that the absolute number is not particularly relevant.

✓ **Author's response:** We sincerely appreciate the reviewer's question. Since the number of samples for each cloud genus is different, the absolute number is not particularly relevant. In this paragraph, we focus more on explaining the reasons for misclassification.

It should be noted that all indices derive from the numbers in the confusion matrices. Therefore, I would present first the matrices, and then the indices, which somewhat summarize what is given in the matrix.

✓ **Author's response:** We sincerely appreciate the reviewer's suggestion. We strongly agree with the reviewer's comment and have adjusted the location of the confusion matrix in the text.

L. 478-491. You don't need to repeat all numbers that are given in the tables. Eventually, you can highlight some numbers in the discussion.

✓ **Author's response:** We sincerely appreciate the reviewer's suggestion. We deleted some content appropriately.

✓ **Modifications: line 478-491:** The comparative analysis results of the above methods are summarized in Table 7. It can be seen from the experimental results that RepVGG had the best performance among the CNN-based methods. Among them, the accuracy rate has the most significant improvement, and the precision and recall rates also have good improvement. The accuracy rate, precison rate, recall rate for the MGCD dataset reached 95.57, 95.31, and 94.99, respectively, while those for the GRSCD dataset were 95.42, 94.99, and 94.88, respectively. Ground-based cloud images have more texture features and deep semantic features than other images, and more image features need to be

obtained to meet the classification requirements of such images. In recent years, Transformer has been widely used for image processing tasks due to its strong feature extraction capability. Several scholars have improved the Transformer derivative model through continuous exploration. Among them, Eff-Swin-T was an improvement based on Swin-T, and its performance on MGCD and GRSCD datasets was better than that of the classic CNN model. Its accuracy rate, precison rate, and recall rate reached 96.93, 96.73, 96.44, and 95.62, 95.41, 95.11, respectively. Compared with Transformer and classical networks, the proposed method had much better classification performance of ground-based cloud images. For different cloud image classification datasets, it exhibited excellent generalization ability and strong robustness, which is instrumental in photovoltaic power generation prediction.

L. 480. You should highlight, at least here (also in the abstract), that the accuracy that you reach is in a classification in 7 classes. There are other papers that use more (and less) cloud categories, so it's important to make sure that the occasional reader knows to what are you referring to.

✓ **Author's response:** We sincerely appreciate the reviewer's suggestion. We emphasized in the abstract and conclusion the accuracy we achieved in classifying across 7 categories, and highlighted in the conclusion the use of cloud categories in other papers.

✓ **Modifications: line 529-531:** In addition, the MGCD and GRSCD ground-based cloud image datasets contain 7 types of cloud categories, which is more than the ground-based cloud image datasets used in other papers. This further demonstrates the excellent performance of the proposed method.

✓ **Modifications: line 23-25:** The proposed method is validated on MGCD and GRSCD ground-based cloud image datasets containing 7 cloud categories, with the respective classification accuracy values of 98.15% and 98.07%, outperforming those of ten most advanced methods used as the reference

English and technical corrections.

✓ **Author's response:** We sincerely appreciate the reviewer's suggestion. We have already found a native English speaker to polish this paper.

Acronyms should be defined the first time they appear both in the abstract and in the text.

✓ **Author's response:** We sincerely appreciate the reviewer's suggestion. We have checked the full text and corrected the placement of acronyms .

L. 10. Clouds impact Earth radiation, not only its "prediction"

✓ **Author's response:** We sincerely appreciate the reviewer's reminder. We strongly agree with the reviewer's thoughts and have revised this sentence.

✓ **Modifications: line 10:**Atmospheric clouds greatly impact the Earth's radiation, hydrological cycle, and climate change.

L. 23. "Accuracy" respect to what? Which is the reference? In other words who or how was the "true" cloud classification established?

✓ **Author's response:** We sincerely appreciate the reviewer's reminder. We apologize for the language error. "Accuracy" refers to "accuracy rate," and we have corrected the sentence.

✓ **Modifications: line 23-25:**The proposed method is validated on MGCD and GRSCD ground-based cloud image datasets containing 7 cloud categories, with the respective classification accuracy rate values of 98.15% and 98.07%, outperforming those of ten most advanced methods used as the reference.

L. 27. "covering" instead of "accounting", I would say.

✓ **Author's response:** We sincerely appreciate the reviewer's suggestion. We strongly agree with the reviewer's suggestion and have revised this sentence.     B bb bnb    nb bv bvb vb v

L. 35. "of the outside to the inside", it should be "from above".

✓ **Author's response:** We sincerely appreciate the reviewer's suggestion. We strongly agree with the reviewer's suggestion and modified it.

L. 47. "budget balance", I think one of the two words is enough.

✓ **Author's response:** We sincerely appreciate the reviewer's suggestion. We strongly agree with the reviewer's suggestion and deleted the word "budget".

L. 70 (and many other places). Do not repeat "Singh et al. (Singh and Glennen, 2005)…"; You can simply write "Singh and Glennen (2005)…"

✓ **Author's response:** We sincerely appreciate the reviewer's suggestion. We have double-checked the entire text and made modifications to the citation format according to the requirements of Atmospheric Measurement Techniques.

L. 344. "diagram" is not the adequate word, in my opinion.

✓ **Author's response:** We sincerely appreciate the reviewer's suggestion. We strongly agree with the reviewer's suggestion and deleted the word "diagram".

L. 430. Is "Ablation" the adequate wording?

✓ **Author's response:** We sincerely appreciate the reviewer's reminder. We have consulted some image classfication paper from the Web of Science and found that many papers use the word "ablation" 。 In addition, we have listed several documents that use the word "ablation" as follows:

[1] Guo, J., Zou, X., Chen, Y., Liu, Y., Hao, J., Liu, J., and Yan, Y.: AsConvSR: Fast and Lightweight Super-Resolution Network with Assembled Convolutions, https://doi.org/10.48550/arXiv.2305.03387, 1 May 2023.

[2] Lu, Z., Wang, Z., Li, X., and Zhang, J.: A Method of Ground-Based Cloud Motion Predict: CCLSTM + SR-Net, Remote Sens., 13, 3876, https://doi.org/10.3390/rs13193876, 2021.

[3] Zhang, L., Wei, W., Qiu, B., Luo, A., Zhang, M., and Li, X.: A Novel Ground-Based Cloud Image Segmentation Method Based on a Multibranch Asymmetric Convolution Module and Attention Mechanism, Remote Sens., 14, 3970, https://doi.org/10.3390/rs14163970, 2022.